# Psychosocial Determinants of Lifestyle Change after a Cancer Diagnosis: A Systematic Review of the Literature

**DOI:** 10.3390/cancers14082026

**Published:** 2022-04-16

**Authors:** Meeke Hoedjes, Inge Nijman, Chris Hinnen

**Affiliations:** 1CoRPS—Center of Research on Psychological and Somatic Disorders, Department of Medical and Clinical Psychology, Tilburg University, 5000 LE Tilburg, The Netherlands; i.l.nijman@tilburguniversity.edu; 2Oncology Center, Leids Universitair Medical Center, 2333 ZA Leiden, The Netherlands; s.c.h.hinnen@lumc.nl

**Keywords:** cancer survivors, lifestyle change, psychological, social, psychosocial, determinants, barriers, facilitators, systematic review

## Abstract

**Simple Summary:**

Although cancer survivors may experience health benefits from favorable lifestyle changes, many cancer survivors do not adhere to lifestyle recommendations or make favorable lifestyle changes after cancer diagnosis. This systematic review of the literature aimed to provide an overview of the scientific literature on sociodemographic, psychological and social determinants that may facilitate or hamper lifestyle change after the diagnosis cancer. It provides a structured overview of the large variety of determinants of changes in different lifestyle behaviors (physical activity, diet, smoking, alcohol, sun protection, and multiple lifestyle behaviors) derived from the 123 included papers (71 quantitative and 52 qualitative). Findings demonstrate the important role of oncology healthcare professionals in promoting healthy lifestyle changes in cancer survivors and inform researchers and healthcare professionals about the methods and strategies they can use to promote healthy lifestyle changes in cancer survivors.

**Abstract:**

The aim of this study is to provide a systematic overview of the scientific literature on sociodemographic, psychological and social determinants that may facilitate or hamper lifestyle change after the diagnosis cancer. Four databases (PubMed, PsychINFO, Cumulative Index to Nursing and Allied Health Literature (CINAHL), and Web of Science) were searched for relevant papers up to October 2021. Of the 9586 references yielded by the literature search, 123 papers were included: 71 quantitative and 52 qualitative papers. Findings showed a large variety of determinants influencing lifestyle change after cancer diagnosis, with differences between lifestyle behaviors (physical activity, diet, smoking, alcohol, sun protection, and multiple lifestyle behaviors) and findings from quantitative vs. qualitative studies. Findings demonstrate the important role of oncology healthcare professionals in promoting healthy lifestyle changes in cancer survivors. In addition, findings inform researchers involved in the development of health promotion programs about the methods and strategies they can use to promote healthy lifestyle changes in cancer survivors. Favorable lifestyle changes are expected to have beneficial effects on cancer risk and overall health in cancer survivors.

## 1. Introduction

A large body of evidence has demonstrated that lifestyle not only influences the risk of developing cancer [1] but also the risk of cancer recurrence, comorbidities such as cardiovascular disease and type II diabetes mellitus, and mortality [2,3,4,5,6]. Moreover, lifestyle has been associated with several biological mechanisms, such as inflammation and Natural Killer cell function, that may impact health-related outcomes [7,8,9,10]. Favorable lifestyle changes, such as increasing physical activity or smoking cessation, may optimize these health outcomes and increase health-related quality of life among cancer survivors (i.e., individuals who are living with a diagnosis of cancer, including those who have recovered from the disease [1]) [11,12,13,14,15,16]. In accordance, lifestyle and body weight recommendations have been issued for cancer survivors, such as the recommendations from the World Cancer Research Fund (WCRF) [1,17]. Despite the potential health benefits, many cancer survivors do not adhere to these recommendations and do not make favorable lifestyle changes after diagnosis [18,19,20,21,22,23]. The reason for this is likely to be complex and multifactorial.

Knowledge on determinants that enhance lifestyle changes (i.e., facilitators) and determinants that limit lifestyle changes (i.e., barriers) in cancer survivors is needed to be able to identify what techniques and strategies can be used to achieve lifestyle changes in this specific patient population. It is important to use behavior change techniques and strategies matching cancer survivor specific determinants, as these are likely to require a (partly) different approach as opposed to other patient populations or the general population. Park & Gaffey (2007) provided an overview of psychosocial determinants of lifestyle change after a cancer diagnosis [24]. Their integrative review included the results of 30 quantitative studies examining relationships among psychosocial factors and lifestyle change in cancer survivors. They concluded that findings of the included studies were inconsistent and that their ability to draw conclusions was limited, predominantly due to mostly cross-sectional study designs and the heterogeneity between the included studies.

To extend the existing literature, this study builds on the review of Park & Gaffey (2007) [24] with an updated, extended, systematic literature search, structured per lifestyle behavior, and additionally including qualitative research. Using both quantitative and qualitative research methods to gain knowledge on determinants, capitalizes the strengths of both research methods [25]. The aim of this study is to provide a systematic overview of the scientific literature on sociodemographic, psychological and social determinants that may facilitate or hamper lifestyle change after the diagnosis cancer. Data on sociodemographic determinants (such as age, gender, educational level, and marital status) may provide insight into which cancer survivors specifically should be targeted to promote lifestyle changes. Data on psychosocial determinants, both at the inter-individual level (determinants at the between-person level, such as social support) and intra-individual level (determinants at the within-person level, such as self-efficacy), provides insight into which modifiable determinants should be targeted for change and informs about what type of techniques or strategies can be used to positively influence these modifiable determinants.

## 2. Materials and Methods

This systematic review of the literature was conducted in line with PRISMA (Preferred Reporting Items for Systematic Reviews and Meta-Analyses) guidelines and was submitted to PROSPERO (International prospective register of systematic reviews; ID313277).

### 2.1. Literature Search

A systematic review of the literature up to the 20 October 2021 was conducted. A total of four databases were searched for relevant papers: (PubMed, PsychINFO, Web of Science, and CINAHL (Cumulative Index to Nursing and Allied Health Literature). A combination of search terms from the following concepts were used: Cancer survivors AND one of the lifestyle behaviors (lifestyle, physical activity, diet, smoking, alcohol consumption, OR sun protection) AND change AND psychosocial (psychological OR social). The complete list of search terms used associated with each concept included in the search is provided in Table 1. When performing the search in the databases a filter for language was applied, including only articles in the English language. Articles retrieved from the database searches were exported to a reference library (EndNote) and combined into one database, in which duplicates were deleted.

### 2.2. Selection Procedure

First, two researchers (IN and MH) simultaneously screened and labelled 10% of the retrieved articles based on title and abstract. Inconsistencies in labelling were resolved by discussion until consensus was achieved. Second, the remaining articles (90%) were divided among these two researchers and were screened and labeled based on title and abstract. After, the full-texts of the articles that were not excluded based on title and abstract were divided among three researchers (IN, CH and MH) and were read and labelled independently to select eligible full-texts. Inconsistencies between the researchers with regard to whether an article should have been included or not were discussed until consensus on inclusion or exclusion was achieved.

Both observational and intervention studies describing psychosocial determinants of change in physical activity, dietary intake, smoking, alcohol consumption, and sun protection among survivors of any type of cancer and any time since diagnosis were included. Articles on other outcome measures (e.g., changes in sleep or weight loss) were excluded. Furthermore, an article was excluded when it described non-human research, when the described study was not original research (e.g., a review article), when the study population did not (only) consist of cancer survivors, when the study did not describe change in one of the lifestyle behaviors of interest, and when the article did not involve psychosocial determinants. 

### 2.3. Data Extraction

The following data were extracted and described separately for each included article: first author and year of publication, country, study design, sample characteristics (sample size, type(s) of cancer, mean age with standard deviation (SD), percentage of female participants, mean time since diagnosis or treatment, and mean baseline Body Mass Index (BMI), psychosocial and lifestyle behavior measurements including measurement instruments, and the findings with regard to psychosocial determinants of lifestyle change. Psychosocial variables were categorized into socio-demographic, inter-individual, and intra-individual determinants.

## 3. Results

In Figure 1, a flow diagram is depicted of inclusion and exclusion of publications derived from the database searches, including reasons for exclusion. In total, the database searches yielded 9586 references. After removal of 2979 duplicates, 6607 titles and abstracts were assessed for eligibility. Of the 176 full-texts that were screened, 123 papers were included: 71 quantitative papers and 52 qualitative papers. See Appendix A for an overview of the characteristics and findings of the included quantitative and qualitative studies.

### 3.1. General Characteristics of the Included Studies 

The majority of the 71 quantitative studies were conducted in the USA (n = 38, 53.5%) [26,27,28,29,30,31,32,33,34,35,36,37,38,39,40,41,42,43,44,45,46,47,48,49,50,51,52,53,54,55,56,57,58,59,60,61,62,63]. The other studies were conducted in Canada [64,65,66,67,68,69,70,71], Australia [72,73,74,75,76,77,78,79], Germany [80,81,82], the Netherlands [83,84], the UK [85,86], Portugal [87], New Zealand [88], Sweden [89], Spain [90], Norway [91], the UK [85,92], Denmark [93], Taiwan [94,95], South Korea [96], Puerto Rico [38], and China [44]. The number of participants in these studies ranged from 23 [40] to 3000 [61]. Median sample size in these 71 quantitative studies was 224.5 (Interquartile Range 354). Most of the studies were conducted in breast cancer survivors (n = 25, 35.2%) [26,28,29,32,35,36,37,38,40,42,43,44,50,51,61,64,65,66,68,70,80,87,90,93,94], five studies were conducted in prostate cancer survivors [73,74,85,89,92], six in colorectal cancer survivors [33,34,49,63,72,96], two in lung cancer survivors [52,95], two in head and neck cancer survivors [55,71], one in in thyroid cancer survivors [79], one in laryngology cancer survivors [54], one in endometrial cancer survivors [27], and one in hematologic cancer survivors [69]. 27 studies (38%) were conducted in survivors of mixed cancer types [30,31,39,41,45,46,47,48,53,56,57,58,59,60,62,67,73,75,76,77,78,81,82,83,84,86,88]. 15 out of 71 studies (i.e., 21.1%) had a cross-sectional study design and assessed changes in lifestyle behaviors following diagnosis and/or treatment retrospectively [26,31,32,62,67,73,76,78,79,83,89,90,91,92,96]. 38 out of 71 studies (i.e., 53.5%) were intervention studies [27,28,33,34,35,36,37,38,39,40,41,42,43,45,46,47,48,50,51,53,54,55,56,57,58,59,60,68,69,70,74,75,77,81,82,84,86,88], of which 35 were randomized controlled intervention studies [28,33,34,35,36,37,38,39,41,42,43,45,46,47,48,50,51,53,54,55,56,57,58,59,60,69,70,74,75,77,81,82,84,86,88], and one was a quasi-randomized controlled intervention study [75]. The duration of these interventions ranged from four weeks [81,82] to four years [51]. 18 out of 71 quantitative studies (i.e., 25.4%) were prospective observational studies [29,30,44,49,52,61,63,64,65,66,71,72,80,85,87,93,94,95]. 

The 52 qualitative studies examining determinants of changes in lifestyle, were conducted in the UK (n = 12) [97,98,99,100,101,102,103,104,105,106,107,108], the USA(n = 10) [109,110,111,112,113,114,115,116,117,118], Canada (n = 10) [119,120,121,122,123,124,125,126,127,128], Australia (n = 8) [129,130,131,132,133,134,135], Sweden (n = 3) [136,137,138], China (n = 2) [139,140], Denmark (n = 2) [141,142], Norway (n = 2) [143,144], France (n = 1) [145], Italy (n = 1) [146], Germany (n = 1) [147], and Taiwan (n = 1) [148]. The number of participants in these studies varied from 4 [128] to 247 [137]. Median sample size in these 52 qualitative studies was 20 (Interquartile Range 15.75). Participants of the qualitative studies were survivors of breast cancer [99,119,121,122,123,125,127,128,135,136,144,145], prostate cancer [102,103,104,106,107,138], colorectal cancer [116,134,139,143], head and neck cancer [126,133], lung cancer [117,124], breast or colon cancer [141], bladder cancer [97], acute leukemia [110], multiple myeloma patients [111], pediatric cancer patients [147], gastrointestinal cancer [98], endometrial cancer [108], adolescent and young adult (AYA) cancer survivors [118], and survivors of mixed types of cancer [100,105,109,115,129,130,140,146,148]. Three studies also included partners [102,104,107], two included caregivers [114,115], and one included supporters [118]. In most of the qualitative studies semi-structured interviews were conducted [98,99,102,103,104,105,106,107,110,111,115,117,119,120,121,122,123,125,127,128,129,130,131,132,133,134,135,138,140,141,142,143,147,148]. In six studies, focus groups were conducted [97,109,126,144,145,146]. In eight studies, both individual interviews and focus groups were conducted [100,108,113,116,118,124,136,139]. In two studies, a mixed-methods design was applied, and both semi-structured interviews and a questionnaire were used to examine determinants of changes in physical activity [101,137]. Most qualitative studies examined lifestyle changes during an intervention [97,98,101,106,107,110,111,112,113,114,117,120,125,135,137,138,139,141,142,143,144,145], and at follow-up after the end of an intervention [98,99,113,128,129]. 10 qualitative studies examined changes after diagnosis [102,103,104,105,115,121,122,123,140,148], four studies examined changes during treatment [127,136,146,147], and 3 studies examined lifestyle changes following cessation of active treatment [108,116,134]. In some studies it was not specified what period the physical activity changes referred to [100,109,118,124,126,130,131,132,133]. Time since diagnosis at the time of the interview ranged from five months [140] to up to 31 years after treatment [119]. 

### 3.2. Psychosocial Determinants

An overview of the psychosocial determinants of lifestyle changes retrieved from the included *quantitative* studies is presented in Table 2. See Table 3 for an overview of the psychosocial determinants of lifestyle changes retrieved from the included *qualitative* studies. Below, both quantitative and qualitative findings on psychosocial determinants of changes in lifestyle are presented per lifestyle behavior.

### 3.3. Changes in Physical Activity

In total, 71 of the included studies described psychosocial determinants of changes in physical activity in cancer survivors (45 quantitative studies and 26 qualitative studies).

#### 3.3.1. Sociodemographic Determinants

12 quantitative studies assessed socio-demographic determinants of changes in physical activity in cancer survivors [28,33,39,44,64,76,78,80,84,85,87,94]. 10 out of those 12 studies assessed *age* as a determinant of change in physical activity [28,33,39,44,64,78,80,85,87,94]. Three out of these 10 studies found an association between age and changes in physical activity. Two found participants with older *age* to be more likely to change towards being physically inactive after diagnosis [39,87], whereas the other study found *age* to predict increased exercise frequency [94]. In the qualitative studies, ageing was reported to be a barrier to increasing physical activity [98,99,130,131,132].

Eight out of the 12 quantitative studies assessing socio-demographic determinants assessed *educational level* as a determinant of changes in physical activity [28,33,44,76,78,80,84,87], of which five did not find a significant association between *educational level* and changes in physical activity [28,33,76,78,80]. The three studies that did find a significant association showed mixed results (see Table 3) [44,84,87]. In the qualitative studies, educational level was not mentioned.

Five quantitative studies assessed *employment status* as potential determinant of changes in physical activity [28,33,78,80,87]. Four out of these five studies did not find a significant association. The one study that did, found increasing sedentary time to be higher in participants working fulltime [33]. In the qualitative studies *work-related factors* (e.g., resuming work, working full-time) were mentioned as barriers to favorable changes in physical activity [99,113,145].

Other socio-demographic determinants that were found to be significantly associated with changes in physical activity were *job position* [80], *social class* [85], and *income* [44]. In two qualitative studies [100,109,119,132,146], financial constraints were mentioned as a barrier to changes in physical activity (e.g., the cost of attending physical activity facilities). On the other hand, affordability, was mentioned as a facilitator [100,109].

*Marital status* [28,78,80,85,87], *race/ethnicity* [28,44], and *years of education* [94] were not found to be associated with changes in physical activity. *Gender* was assessed as a potential determinant in two quantitative studies that found mixed results [33,39]. See Table 2. Gender was not specifically mentioned as a determinant in the qualitative studies.

In the qualitative studies, *poor weather conditions* was frequently mentioned as a barrier to changes in physical activity [97,98,99,109,112,113,114,119,129,131,146,147]. *Environmental factors, such as poor infrastructure, geographical isolation, and lack of footpaths*, were also mentioned as barriers in qualitative studies [114,129,132], whereas a *pleasant local physical activity environment* was mentioned as a facilitator of physical activity changes [132]. See Table 3. 

#### 3.3.2. Inter-Individual Determinants

11 quantitative studies assessed inter-individual determinants of changes in physical activity [26,28,34,35,38,43,44,45,81,94]. 10 of those studies assessed *social support* as a potential determinant of changes in physical activity [26,28,29,34,35,38,43,44,45,94]. Seven out of these 10 studies found *social support* to be significantly positively associated with changes in physical activity [26,28,34,35,38,44,45,94]. In these studies, whom offered social support and when changes occurred differed. See Table 2. Social support from partners and family members was frequently mentioned as a facilitator of physical activity changes in qualitative studies [97,98,100,111,113,114,119,120,129,130,131,132,136,137,141,144,145,146], whereas *social isolation* was mentioned as a barrier [129,147]. Also, *receiving advice or support from health care professionals* [101,130,131,141,144,146] and *receiving professional supervision* [97,98,100,110,137,141,142,143,144,145,146] were mentioned as facilitators. In addition, the *benefits of exercising with fellow sufferers* was mentioned as a facilitator of increasing physical activity [97,113,120,136,137,141,142]. See Table 3. 

Frequently mentioned barriers of physical activity at the inter-individual level in the qualitative studies include *lack of information or advice from health care professionals* [100,109,119,131,132], *competing time demands* (e.g., competing family or work demands, balancing motherhood with exercising) [98,99,109,112,114,119,120,129,131,132,136,144,145,146,147], and *issues with facilities or resources* (e.g., proximity/access to facilities) [99,109,113,119,120,130,132]. Another frequently mentioned facilitator included *routine and structure* (e.g., having scheduled appointments for exercise) [110,136,137,143,144,145,146]. See Table 3. 

See Table 2 for an overview of the results of the few quantitative studies assessing inter-individual determinants other than social support (*role models* [43,81], *social modeling* [38]).

#### 3.3.3. Intra-Individual Determinants

39 quantitative studies assessed intra-individual determinants of change in physical activity [26,27,29,30,31,33,34,35,36,37,38,39,40,41,42,43,44,45,46,64,65,66,67,68,69,72,73,74,75,76,77,82,83,86,87,88,89,94,96]. 16 studies assessed *self-efficacy* as potential determinant of changes in physical activity [27,30,33,34,35,37,38,39,40,45,68,73,76,86,88,94]. Nine out of these 16 studies found *self-efficacy* to be a significant determinant of change in physical activity [27,30,34,35,37,38,39,73,88], with higher *self-efficacy* to be associated with a greater increase in physical activity [27,30,35,88], greater physical activity adoption and maintenance [34], and greater odds of being sufficiently active at follow-up [37]. Also, decreasers of physical activity reported lower *self-efficacy* than increasers and maintainers [73], and lower *self-efficacy* was more prevalent in physical activity trajectories with the lowest amount of physical activity over time [39]. Besides the general concept of self-efficacy, different types of self-efficacy were assessed (*barriers self-efficacy* [27,42,43,74,75], *task self-efficacy* [42,43,74], *maintenance self-efficacy* [77,82], and *relapse self-efficacy* [75]). Overall, results for these specific types of *self-efficacy* were in line with the results for *self-efficacy*, showing a positive association between types of *self-efficacy* and favorable changes in physical activity [42,43,74,75,77,82]. See Table 2. In qualitative studies, *self-efficacy* was mentioned as facilitator of changes in physical activity [101,111,129,132,141,144], whereas *low self-efficacy* was mentioned as a barrier [100,111,119]. 

In the qualitative studies, besides *physical complaints/physical side effects of treatment* [98,99,100,101,109,110,111,112,114,119,120,129,130,131,132,136,137,141,142,143,144,145,146,147], *psychological complaints (e.g., depression, anxiety, stress)* [98,100,101,109,110,112,114,129,136,147] were frequently mentioned as a barrier to changes in physical activity. Of the quantitative studies, six examined *depressive symptoms* as potential determinants of changes in physical activity [26,29,44,64,72,87]. Five of these studies found *depressive symptoms* not to be significantly associated with changes in physical activity [26,29,44,72,87], whereas one study found participants with higher levels of *depressive symptoms* were significantly less likely to remain sufficiently active [64]. Of the two quantitative studies examining *anxiety* [72,87], one found that participants with higher *anxiety* were significantly less likely to increase their physical activity [72]. See Table 2 for the results of the few quantitative studies per emotional factor (e.g., *emotional distress, fear of cancer recurrence, cancer specific concern, mental health status, shame, and guilt*) on the association with changes in physical activity, which were generally inconsistent or inconclusive [26,31,33,41,44,64,66,83,94]. In the qualitative studies, *concerns and anxiety about exercising* [99,100,109,137,145,147] and *concerns and fears related to symptoms (e.g., body esteem, colostomy bag leakage)* [97,98] were mentioned as barriers. 

Cognitive and behavioral factors were also mentioned as determinants of changes in physical activity. Of the four studies examining *cognitive and behavioral processes* [33,34,37,40], three studies found a significant positive association with favorable changes in physical activity [34,37,40] and two studies found conflicting associations [33,34]. Two quantitative studies examining *action planning* both found a significant association with changes in physical activity [77,96], but not for *coping planning* [96]. In the qualitative studies, *action planning and goal setting* was mentioned as a facilitator [101,114,132,146]. In one quantitative study, *goal setting* was associated with changes in physical activity [74]. Also, *(self-)monitoring and feedback on behavior* [97,98,101,132,137,142,143,144,145] was frequently mentioned as a facilitator in the qualitative studies. Two of the three quantitative studies examining *decisional balance* found it not to be significantly associated with changes in physical activity [34,37], whereas the other study found *decisional balance* to be associated with physical activity adoption, but not maintenance [34]. Also, a higher *stage of change* (a higher readiness to change) was found to be a significant positive predictor of change in physical activity [76]. 

Determinants related to motivation for physical activity changes were examined in four quantitative studies. Two quantitative studies assessing *motivation* found conflicting results [74,75]. One study assessing *motivational regulations* found changes in *self-determined motivation* to be positively related to changes in physical activity [65]. Another quantitative study examining *motivational processes* found that perceived opportunity was a significant mediator of exercise behavior [69]. In the qualitative studies, *personal and/or internal motivation* was mentioned as a facilitator [101,113,137,141,143], whereas *lack of motivation* [98,99,101,109,112,114,119,129,130,131,132,145,146,147] was mentioned as a barrier for changes in physical activity. 

The included studies reported on the relation between perceptions or expectations and changes in physical activity. Four quantitative studies assessed *perceived barriers*, of which two found no association with changes in physical activity [73,74], and the other two showed mixed results [42,94]. Six quantitative studies examined *outcome expectations* as a determinant of change in physical activity [27,30,42,43,74,94], of which two found a positive association [74,94]. *Perceptions of physical activity* improving quality of life and overall survival was found to be associated with increased physical activity [67], while *exercise beliefs of negative impact of exercise on cancer* was found to be associated with decreased physical activity [73]. In the qualitative studies, *perceived or anticipated benefits* of lifestyle change (e.g., to improve health, wellbeing, reduce symptoms, improving treatment efficacy & cancer prognosis) were mentioned as facilitators [98,101,111,119,132,144,147]. 

Furthermore, *experienced benefits* from physical activity (e.g., *improving mental wellbeing, processing negative thoughts and feelings*) [97,98,100,101,110,111,114,119,120,129,132,136,137,141,142,143,144,145,146,147] were frequently mentioned as facilitators in the qualitative studies. Another frequently mentioned facilitator of physical activity changes mentioned in the qualitative studies was *enjoyment of being physically active* [98,109,113,119,120,141,146], whereas *lack of enjoyment of physical activity* [98,99,132,137,143] and *not being the sporty type* [99,101,114,131,145,146] were mentioned as barriers. One of the two quantitative studies on *physical activity enjoyment* found no significant association [43], the other found that an increase in *physical activity enjoyment* significantly predicted physical activity at post-intervention [82]. Another frequently mentioned facilitator in the qualitative studies was the perception that being more physically active was experienced as a way of *being able to do something and re-gain control over their lives* [97,98,100,113,120,136,144,146]. 

Four quantitative studies assessed *fatigue* as potential determinant [26,64,68,94]. Three studies found *fatigue* to be a significant determinant of changes in physical activity [26,64,68], with less *fatigue* being associated with increased physical activity [26], *fatigue* being a significant predictor of physical activity maintenance [68], and participants with higher levels of *fatigue* were less likely to remain consistently sufficiently active [64]. 

Other intra-individual determinants that were found to be statistically significantly associated with changes in physical activity in one or two quantitative studies were *health-related quality of life (HRQoL)* [29,89], *intention* [77], *perceived access to exercise* [35], *somatization*) [72], *illness perceptions* [73], *illness representations* [76], and *self-leadership* [96]. See Table 2.

### 3.4. Dietary Changes 

30 studies reported on psychosocial determinants of dietary changes (21 quantitative studies and nine qualitative studies).

#### 3.4.1. Socio-Demographic Determinants

Nine studies assessed socio-demographic determinants of dietary changes in cancer survivors [61,70,76,78,80,87,90,91,92]. Seven out of those nine studies assessed *age* as a determinant of lifestyle changes [61,70,78,80,87,90,91]. Of the four studies that did find an association between *age* and dietary changes [70,78,90,91], three found that younger cancer survivors were more likely to make favorable dietary changes [70,90,91] and one found that older cancer survivors were more likely to make favorable lifestyle changes [78]. Ageing was not mentioned as a determinant of dietary changes in the qualitative studies. 

Eight studies assessed *educational level* as a potential determinant of dietary changes [61,70,76,78,80,87,91,92]. Six out of these eight studies did not find a statistically significant association between educational level and dietary changes [70,76,78,80,87,91], while two studies found that a higher level of education was associated with making favorable dietary changes [61,92]. All four quantitative studies assessing associations between *marital status* and dietary changes found no statistically significant associations [78,80,87,91].

See Table 2 for the results of the socio-demographic determinants that were assessed by one or two quantitative studies (e.g., *employment status* [78,80], *job position* [80], *income* [44,70], *social class* [90], *and cohabitation* [90]). Other socio-economic determinants for dietary changes in the qualitative studies included *financial constraints* (e.g., not being able to afford healthy products) [123,138], and *work-related factors* (e.g., shift work, being retired) [123]. 

#### 3.4.2. Inter-Individual Determinants

Although marital status specifically was not mentioned in the qualitative studies, *social support from family, friends, and health care professionals* was frequently mentioned as a facilitator of dietary changes [102,103,104,122,123,138,139]. In three of the quantitative studies, *social support* was assessed as a potential determinant of dietary changes [26,35,44]. Two out of these three studies found that social support determined favorable dietary changes [35,44]. In the qualitative studies, many other inter-individual determinants were reported (see Table 3), such as *lack of information or advice from health-care professionals* as a barrier to dietary changes [102,103,138].

#### 3.4.3. Intra-Individual Determinants

Of the quantitative studies, 17 assessed intra-individual determinants of dietary changes [26,31,35,36,44,46,47,48,49,50,51,70,76,83,87,89,92]. Six of these studies assessed *depressive symptoms* as determinant of dietary changes, and of these, one study found that depression was a barrier [32]. The other quantitative studies did not find a statistical significant association between depressive symptoms and dietary changes [26,44,50,51,87]. In the qualitative studies, *depressive symptoms* were not mentioned as a barrier to dietary changes. 

Of the five quantitative studies assessing the association between *self-efficacy* and dietary changes [35,47,48,50,76], three found statistically significant associations indicating that higher self-efficacy was associated with favorable dietary changes [35,47,48]. In the qualitative studies, self-efficacy was not mentioned as a determinant of dietary changes. 

Four quantitative studies examined stress-related variables: *stressful life events* [26,70], *contemporary life stress* [36], *psychological distress at diagnosis* [70], and *cancer-related stress* [49]. Although one study found that a greater number of stressful events in the five years preceding diagnosis was associated with initiating dietary change [70], other studies found no statistically significant association between stressful life events [26] or contemporary life stress [36] and dietary changes. One study found that higher initial *psychological distress* at diagnosis was associated with initiating dietary change [70]. Another study found that *cancer-related stress* was a barrier to fruit and vegetable consumption around the diagnosis, but facilitated positive dietary changes by the end of the first year after diagnosis [49]. In the qualitative studies, stress-related variables were not specifically mentioned as determinants of dietary changes.

See Table 2 for the results on the intra-individual determinants of dietary changes examined by one or two quantitative studies, such as *perceived barriers*, *health-related quality of life*, *fear of recurrence, stage of change illness representations*, *perceived behavioural control*, *dispositional optimism*, and *cancer coping style* [26,31,44,46,48,50,76,87,89,92,102,103,121,123,139].

Frequently mentioned barriers to dietary changes in the qualitative studies that were not assessed in the quantitative studies include *perceived/anticipated benefits of lifestyle change* (e.g., to improve health, wellbeing, reduce symptoms, improving treatment efficacy & cancer prognosis) [102,115,121,138] and *lifestyle change as active coping strategy: doing something to gain a sense of control* [102,103,115,121,138]. See Table 3 for an overview of the determinants of dietary change mentioned in the qualitative studies. 

### 3.5. Changes in Smoking Behavior

16 studies described psychosocial determinants of changes in smoking behavior, of which 12 were quantitative and four were qualitative studies.

#### 3.5.1. Sociodemographic Determinants

Eight quantitative studies assessed socio-demographic determinants of changes in smoking behavior in cancer survivors [52,53,55,56,59,60,93,95]. Seven out of those eight studies assessed *age* as a determinant of changes in smoking behavior [52,53,55,56,59,60,95]. Four out of those seven studies did not find a significant association between *age* and changes in smoking behavior [53,55,59,60]. The other three studies that did find an association between *age* and changes in smoking behavior found that older participants were more likely to have been abstinent from smoking [52,56,95]. Age was not mentioned as a determinant of changes in smoking behavior in the qualitative studies. 

Six out of the eight quantitative studies assessed *educational level* as a determinant of change in smoking behavior [52,53,59,60,93,95]. Five out of these six studies did not find a statistically significant association between *educational level* and changes in smoking [52,59,60,93,95]. The one study that did find a significant association between *educational level* and changes in smoking found that long-term cessation rates were lower among those with lower *educational levels* [53]. Educational level was not mentioned as a determinant of changes in smoking behavior in the qualitative studies. 

Of the five quantitative studies assessing *marital status* as a determinant of change in smoking behavior [52,53,59,93,95], one found a marginally significant association between *marital status* and changes in smoking behavior, with married participants yielding higher abstinence rates in the intervention group [59]. Marital status was not explicitly mentioned as a determinant in the qualitative studies. 

Six quantitative studies assessed *gender* as a predictor of changes in smoking behavior [53,55,56,59,60,95]. Whereas five of these studies did not find a significant association [53,55,59,60,95], one study found that participants were more likely to have been abstinent at one of the follow-up measurements if they were male [56]. The four quantitative studies assessing *race* [53,55,59,60] found no statistically significant associations.

The two quantitative studies assessing *income* [53,95] as potential determinant of changes in smoking behavior found no statistically significant associations. In the qualitative studies, *lack of work (e.g., being unemployed or not able to work after cancer diagnosis)* was mentioned as a barrier to smoking cessation [133]. Also, *affordability and smoking cessation saving money* were mentioned as facilitators of smoking cessation in the qualitative studies [105,140]. 

Two quantitative studies examined *second-hand smoke exposure at home* [52,95], of which one study found that being exposed to second-hand smoking at home was significantly associated with being indecisive for abstinence [95]. The other study did find a significant association between *having household members that smoke* and continued smoking univariately, which only remained marginally significant when examined multivariably [52].

#### 3.5.2. Inter-Individual Determinants

Two quantitative studies examined inter-individual determinants of changes in smoking behavior [55,71]. One study did not find *social support* to be a significant predictor of smoking cessation [71], whereas the other study did find significant differences between continuous abstainers and participants that relapsed in some, but not all, supportive behaviors [55]. One study assessed *social smoking environment* as possible determinant and found that participants were more likely to quit smoking if they had a spouse who did not smoke, and fewer peers who smoked [71].

In the qualitative studies, *social support (e.g., from partners and family members)* [124,133,140], *advice or support from health care professionals* [124,133], and *the social unacceptability of smoking* [105,124] were mentioned as facilitators of favorable changes in smoking behavior. *Lack of discussion about lifestyle with health care professionals* [105,124] was mentioned as a barrier to favorable changes in smoking behavior. See Table 3 for an overview of all determinants at the inter-individual level retrieved from the qualitative studies.

#### 3.5.3. Intra-Individual Determinants

Nine quantitative studies assessed intra-individual determinants of changes in smoking behavior [52,53,54,55,57,58,59,60,95]. Quantitative studies assessing *emotional* or *psychological distress, stress coping, and perceived stress* as a determinant found no significant associations with abstinence [55,58,60]. In qualitative studies, the *stress of being away from home (while in hospital)* [133], *psychological complaints* [133], and *coping with emotional distress trough unhealthy behaviors* [105] were mentioned as barriers to favorable changes in smoking behavior.

Of the six quantitative studies assessing *depression* as a determinant of changes in smoking behavior, three did not find a significant association [55,59,95]. The other three studies did find *depression* to be a significant predictor of changes in smoking behavior, with *depression* being associated with continued smoking [52], relapse after quitting [53], and lower abstinence rates [57]. One of the three quantitative studies examining *anxiety* as a potential determinant of change in smoking behavior found that lower levels of *anxiety* significantly predicted abstinence [60]. The other two studies did not find significant associations between *anxiety* and change in smoking behavior [55,95].

Four quantitative studies assessed whether *self-efficacy* was a determinant of changes in smoking behavior [53,55,59,95], of which three studies found that long-term cessation [53] and perseverance for abstinence [95] were less likely among participants with lower *self-efficacy*, and that relapsers expressed significantly lower levels of confidence in their ability to stay off cigarettes [55].

Four quantitative studies assessed *stages of change* [53,54,55,58]. Of the three studies that found significant associations, one found a relationship between *stage of change* and long term smoking status [53], one study found that participants with a higher *stage of change* were more likely to quit smoking [58], and one study found *stage of change* to significantly differentiate between continuous abstainers and relapsers, with the higher the *stage of change*, the less likely the patient was to relapse [55].

Stage of change, self-efficacy, and risk perception were not mentioned as a determinant in the qualitative studies. In contrast, *lack of knowledge and limited perceptions on smoking cessation and health consequences* [105,140], *not perceiving any benefits of smoking cessation* [140], and *not being too concerned about effects of smoking* [133] were mentioned as barriers to favorable changes in smoking behavior in the qualitative studies. See Table 3 for an overview of the intra-individual determinants mentioned in the qualitative studies.

See Table 3 for the results on the intra-individual determinants that were assessed by one or two studies, such as *risk perception* [58,59], *fatalism* [58], *fear of cancer recurrence* [59], *pain* [59], *anger* [55], *confusion* [55], *fatigue* [55,59], *vigor* [55], *pros and cons of quitting* [58].

### 3.6. Changes in Alcohol Consumption

Four quantitative studies reported on determinants of changes in alcohol consumption.

#### 3.6.1. Socio-Demographic Determinants

Three quantitative studies assessed socio-demographic determinants of changes in alcohol consumption in cancer survivors [61,85,93]. The two studies assessing *educational level* as a potential determinant of change in alcohol consumption found mixed results [61,93]. The studies assessing *marital status* [85,93], *age* [61,85], and *social class* [85] as potential determinants of change in alcohol consumption did not find significant associations.

#### 3.6.2. Inter-Individual Determinants

The only quantitative study assessing an inter-individual determinant of changes in alcohol consumption found that *social support* was not a significantly associated with changes in alcohol consumption [61].

#### 3.6.3. Intra-Individual Determinants

Two studies assessed intra-individual determinants of changes in alcohol consumption [31,61]. Higher *fear of cancer recurrence* and higher *emotional distress* were found to be significantly associated with increased alcohol consumption [31]. *Depressive symptoms* and *dispositional optimism* were not found to be significantly associated with changes in alcohol consumption [61].

### 3.7. Changes in Multiple Health Behaviors

17 studies reported on psychosocial determinants of changes in multiple lifestyle behaviors (13 qualitative papers) or a lifestyle score (four quantitative papers).

#### 3.7.1. Socio-Demographic Determinants

Three of the four quantitative studies assessed socio-demographic determinants of changes in lifestyle scores consisting of a combination of multiple health behaviors [32,62,79]. All three assessed *age* as a determinant of change and found no significant associations with any of the lifestyle scores [32,62,79].

In one qualitative study, *ageing* was mentioned both as barrier (e.g., viewing themselves as too old for playing sports) and facilitator (e.g., heightened awareness of susceptibility to illness due to ageing) of lifestyle change [106]. Of the three quantitative studies assessing *educational level* as a potential determinant, two found no significant associations with change in lifestyle behaviors (sleep, diet, exercise, and stress management) or change in substance use (alcohol and smoking) [79], or change in diet or physical activity [62]. The other study found participants with a higher *level of education* to be more likely to make positive changes in physical activity or diet [32]. One study assessed *gender* as a determinant and found female *gender* to be significantly related to less positive change in substance use (smoking and alcohol consumption), but not to be related to change in lifestyle behavior (sleep, diet, physical activity, and stress management) [79]. Other socio-demographic determinants that were not found to be significantly associated with lifestyle behavior changes were *marital status* [62,79], *employment* [79], *income* [79], and *race* [32,62]. See Table 2. In the qualitative studies, *poor weather conditions* [107,108,125], *financial constraints* [106,108,118] and *environmental factors (such as poor infrastructure)* [108] were mentioned as barriers to lifestyle changes, while *environmental factors (e.g., proper infrastructure)* [108] and *good weather* [108] were mentioned as facilitators.

#### 3.7.2. Inter-Individual Determinants

Only one quantitative study assessed inter-individual determinants of changes in lifestyle behaviors [62]. This study found that *social support* was a significant predictor of positive behavior change (physical activity and diet), whereas no significant associations were found with social constraints [62]. In many qualitative studies, *social support from partners and family members* [106,107,108,117,118,125,126,135] and *advice or support from health-care professionals* [106,117,125,126,128,135] were mentioned as facilitators for lifestyle changes, whereas *lack of information or advice from health care professionals* [106,108,118,126,135,148], *poor support and understanding from family members* [135], and *living alone or not having a partner* [107,117,125,127] were mentioned as barriers of lifestyle change. See Table 3 for an overview of all reported determinants of lifestyle change in the qualitative studies.

#### 3.7.3. Intra-Individual Determinants

Three quantitative studies assessed intra-individual determinants of lifestyle change [62,63,79]. Two of those three studies examined *cancer-related (dis)stress* as a potential determinant. One study did not find cancer-related stress to be associated with changes in lifestyle behavior [79], whereas the other study examined two subscales of *cancer-related distress (intrusions and avoidance)* and found only cancer-related intrusions to be a significant predictor of positive behavior change [62]. In addition, one study found that an increase in *anxiety* symptoms was related to greater odds of reporting an unhealthy lifestyle (physical activity, diet, BMI, alcohol and tobacco consumption) [63]. Other intra-individual determinants found not to be significant determinants of changes in lifestyle behaviors were *depression* [63], and *traumatic stressor response* [62]. See Table 2.

In the qualitative studies, *concerns or fears related to symptoms (e.g., colostomy bag leakage and accidents)* [107,134], *coping with (emotional dis)stress through unhealthy behaviors* [126,127,135], and *psychological complaints such as low mood, depression, stress and anxiety* [108,118,126,127] were reported as perceived barriers for lifestyle changes. On the other hand, *fear of recurrence* and *perceiving that lifestyle change may prevent recurrence* [118,126,128,135,148] was mentioned as a facilitator of lifestyle changes.

One quantitative study found *benefit finding* to be associated with a significant increase in lifestyle behavior (sleep, diet, physical activity, and stress management), but not with substance use (alcohol consumption and smoking) [79]. Another quantitative study examining *optimism* found it to be a significant predictor of positive lifestyle behavior change (diet and physical activity) [62].

In the qualitative studies, after *treatment side effects* [106,107,108,116,118,125,126,127,135], *perceiving no need for lifestyle change* [106,108,116], *beliefs about (the cause of) cancer being unrelated to lifestyle* [106,116,126], *low self-efficacy* [116,128,134], *not enjoying healthy behaviors* [107,108,125], and *uncertainty about benefits of lifestyle in relation to cancer and health* or *not perceiving any benefits of lifestyle change* [106,116,126,134] were most often mentioned as intra-individual barriers to lifestyle change. On the other hand, in the qualitative studies, the following factors were most frequently mentioned as facilitators at the intra-individual level: *cancer diagnosis as wake up call, as initial motivating factor* [106,107,108,117,126], *perceived/anticipated benefits of lifestyle change* (e.g., to improve health, wellbeing, reduce symptoms, improving treatment efficacy & cancer prognosis) [106,107,108,116,118,148], *experienced benefits from healthy behaviors* (e.g., improved mental wellbeing; help process negative thoughts and feelings) [106,108,116,128,135,148], *personal/internal motivation and commitment* [107,117,134,135], and *goal setting/action planning* [108,117,118,125,128]. See Table 3 for an overview of all barriers and facilitators of lifestyle change retrieved from the qualitative studies.

### 3.8. Changes in Sun Protection Behavior

Two quantitative studies reported on determinants of changes in sun protection behavior.

#### 3.8.1. Socio-Demographic Determinants

One of the two studies assessed socio-demographic determinants of changes in sun protection in cancer survivors [78]. Being *older* than 55 years was found to be significantly associated with increased sun protection behavior as compared to being younger than 55 years [78]. *Marital status*, *employment status*, and *educational level* were not found to be significant predictors of changes in sun protection behavior [78].

#### 3.8.2. Inter-Individual Determinants

None of the studies examined inter-individual determinants of changes in sun protection behavior.

#### 3.8.3. Intra-Individual Determinants

One study assessed intra-individual determinants of changes in sun protection behavior [31]. *Fear of cancer recurrence* and *emotional distress* were both not found to be significantly associated with changes in sun protection behavior [31].

## 4. Discussion

This systematic review of the literature on psychosocial determinants of lifestyle changes in cancer survivors provides a broad and structured overview of psychosocial determinants per lifestyle behavior on the socio-demographic, inter-individual, and intra-individual level retrieved from both quantitative and qualitative research. To our knowledge, this is the first review on psychosocial determinants of lifestyle change in cancer survivors including qualitative research.

Of the quantitative studies assessing sociodemographic determinants, most assessed *educational level* as potential determinant of lifestyle change in cancer survivors. These studies mostly showed no association between educational level and lifestyle change [28,33,44,52,59,60,62,70,76,78,79,80,87,91,93,94,95]. The studies that did find a statistically significant association showed that higher educational level was associated with more favorable lifestyle changes [32,44,61,84,87,92]. *Age* and *marital status* were the next most frequently assessed socio-demographic determinants of lifestyle change in the quantitative studies. These studies showed that *marital status* was not associated [28,52,53,62,78,79,80,85,87,91,93,95], as did most of the studies assessing *age* [28,32,33,44,53,55,59,60,61,62,64,78,79,80,85,87]. Ten out of the 35 studies (i.e., 28.6%) assessing age did find an association between age and lifestyle changes. For example, the studies that did find an association between age and smoking behavior suggested older age to be associated with favorable changes in smoking behavior [52,56,95]. In the qualitative studies, *ageing* was reported as a barrier to being more physically active [98,99,130,131,132]. Besides ageing, different determinants of lifestyle changes at the sociodemographic level were mentioned in the qualitative studies. Overall, sociodemographic factors were more frequently mentioned as barriers than as facilitators in the qualitative studies. Most qualitative studies mentioned *poor weather conditions* as a barrier to being more physically active [97,98,99,109,112,113,114,119,129,131,146,147]. Also, *financial constraints* (e.g., healthy products being more expensive, costs of using exercise facilities) were mentioned as a barrier to making favorable lifestyle changes [100,106,108,109,118,119,123,132,138,146], while *affordability* of making lifestyle changes or the financial benefit of smoking cessation was mentioned as a facilitator [100,105,109,140]. *Environmental factors* (e.g., geographical isolation, lack of footpaths) [108,114,129,132] and *work-related factors* (e.g., working full-time) [99,113,123,133,145] were also mentioned as socio-demographic determinants in the qualitative studies, primarily as barriers. Overall, our results are in line with and build upon the relatively few studies examining socio-demographic determinants in the review of Park & Gaffey (2007) [24]. As in our study, Park & Gaffey (2007) found that marital status was not associated with lifestyle changes and that the relationship with age and educational level was inconsistent. A systematic review by Kampshoff et al. (2014) [149] examining determinants of physical activity maintenance in cancer survivors, found similar results with no association with marital status, and inconsistent results regarding age and educational level.

Most quantitative studies examining inter-individual determinants of lifestyle changes, assessed associations between *social support* and lifestyle changes. Although six of these studies did not find a significant association [26,43,44,55,61,71], the nine studies that did find an association showed a positive association between social support and favorable lifestyle changes, particularly in physical activity [28,29,38,63,71,94] and diet [35,44]. In the qualitative studies, *social support (e.g., from partner and family members)* was the most frequently mentioned facilitator of favorable lifestyle changes. It was mentioned in 34 of the 52 (i.e., 65.4%) included qualitative studies [97,98,100,102,103,106,107,108,111,113,114,117,118,119,120,122,123,124,125,126,129,130,131,132,133,135,136,137,138,140,141,144,145,146]. Apart from social support, many other inter-individual determinants of lifestyle changes were mentioned in the qualitative studies. The most frequently mentioned inter-individual determinants in the qualitative studies included *advice or support from health care professionals* [101,102,104,106,117,124,125,126,128,130,131,133,135,138,139,141,144,146] and *receiving professional supervision* [97,98,100,110,133,137,141,142,143,144,145,146], which were mentioned as facilitators of favorable lifestyle changes, whereas *lack of information or advice from health care professionals* was mentioned as a barrier to making favorable lifestyle changes [100,102,103,106,108,109,118,119,126,131,132,135,138,148]. Another frequently mentioned barrier in qualitative studies was *competing time demands* (e.g., competing work or family demands) [98,99,105,107,108,109,112,114,119,120,125,128,129,131,132,136,144,145,146,147]. The review by Park & Gaffey (2007) [24] found mixed results across lifestyle behaviors regarding the association between social support and lifestyle change, with social support being related to increased exercise, and abstinence from smoking, but no studies showing social support to be related to making dietary changes.

Of the quantitative studies assessing determinants of lifestyle changes in cancer survivors at the intra-individual level, *self-efficacy* was by far the most studied. In those studies, some form of self-efficacy (*self-efficacy* [27,30,33,34,35,37,38,39,40,42,45,47,48,50,53,55,58,59,68,73,76,77,86,88,94,95], *task self-efficacy* [42,43], *barriers self-efficacy* [27,38,43,74,75], *relapse self-efficacy* [75], *and maintenance self-efficacy* [77,82]) was assessed. The vast majority of these studies, assessed associations between self-efficacy and changes in physical activity. More than half of these studies found an association between higher levels of self-efficacy and favorable lifestyle changes. In the qualitative studies, *self-efficacy* [101,111,116,129,132,135,141,144] was mentioned as a facilitator while *low self-efficacy* was mentioned as a barrier [100,111,116,119,128,134], primarily of changes in physical activity. Similarly, half of the studies included by Kampshoff et al. (2014) [149] found a positive association between self-efficacy and maintenance of physical activity in cancer survivors, whereas the other half of the studies found no significant associations. In the review by Park & Gaffey (2007) [24] less studies investigated self-efficacy as a potential determinant of lifestyle changes and these studies found mixed results across lifestyle behaviors.

The second most studied intra-individual determinant of lifestyle changes in the quantitative studies was *depressive symptoms*. Of the studies in which depressive symptoms were assessed as potential determinant of lifestyle changes in cancer survivors, most studies did not find a statistically significant association [26,29,44,50,55,59,61,63,72,87,95]. The studies that did find a significant association found that higher levels of depressive symptoms were associated with unfavorable changes in physical activity [26,64], diet [51], and smoking [52,53,57]. These findings extend the findings of the fewer studies included in the review of Park & Gaffey (2007), generally suggesting that cancer-related distress was associated with favorable changes in lifestyle behavior, as in the broader literature on lifestyle change [24]. In the qualitative studies, besides *physical complaints or treatment side-effects,* which were most frequently mentioned as barrier to lifestyle changes (primarily in physical activity) at the intra-individual level [98,99,100,101,106,107,108,109,110,111,112,114,116,118,119,120,125,126,127,129,130,131,132,135,136,137,141,142,143,144,145,146,147], *psychological complaints (e.g., depression, anxiety, and stress)* were frequently mentioned as barriers to lifestyle changes [98,100,101,108,109,110,112,114,118,126,127,129,133,136,147].

The third most studied potential determinants of lifestyle changes at the intra-individual level in the quantitative studies were *anxiety* [50,55,60,63,72,87,95] and *stages of change* [50,53,54,55,58,76]. Of the quantitative studies that assessed anxiety, four out of seven did not find a statistically significant association with lifestyle changes [50,55,87,95]. The other three studies found an inverse relationship between symptoms of anxiety and favorable lifestyle changes [60,63,72]. Similar results were reported by Park & Gaffey (2027) [24]. Kampshoff et al. (2014) [149] also found anxiety not to be related to maintenance of physical activity. In the qualitative studies, anxiety was also mentioned as a barrier to lifestyle changes as part of the *psychological complaints* cancer survivors experienced after diagnosis, but also as *anxiety specifically related to exercising* [99,100,109,137,145,147] and *fears related to symptoms* [97,98,107,134], while *fear of recurrence and the perception that lifestyle change may prevent recurrence* was mentioned as a facilitator of lifestyle changes [102,103,113,118,121,123,126,128,135,139,148].

Of the six quantitative studies that assessed *stages of change*, four found a statistical significant association between a higher stage of change and favorable lifestyle changes, mostly in smoking behavior [53,55,58,76], and one found a borderline significant association [54]. The two studies examining stage of change in the review by Park & Gaffey (2007) [24] found higher stage of change to be related to continued abstinence of smoking and increased physical activity. Stage of change was not mentioned in the qualitative studies.

A frequently mentioned barrier to making lifestyle changes in the qualitative studies was *lack of motivation* (n = 17) [98,99,101,109,112,114,119,129,130,131,132,134,135,138,145,146,147]. Contrary, *personal, internal motivation and commitment* was mentioned as a facilitator of lifestyle changes [101,107,113,117,122,134,135,137,139,141,143]. In addition, perceiving the *cancer diagnosis as a wake-up call or initial motivating factor* was mentioned as a facilitator for lifestyle changes in qualitative studies [102,106,107,108,117,122,126,133]. Motivation was assessed as determinant of lifestyle changes in four quantitative studies, only for changes in physical activity [74,75]. Findings of these studies were inconsistent, with two studies suggesting a positive association between motivation and favorable changes in physical activity [65,69]. In the review by Park & Gaffey (2007) [24], only one study examining motivation was reported, which findings showed an inverse relation to smoking and alcohol consumption.

The most frequently mentioned facilitator of lifestyle changes in the qualitative studies at the intra-individual level was *experienced benefits from healthy*
*behaviors* [97,98,100,101,103,106,108,110,111,114,116,119,120,128,129,132,135,136,137,139,141,142,143,144,145,146,147,148]. The next most frequently mentioned facilitator was *perceived or anticipated benefits of lifestyle change* (e.g., to improve health, wellbeing, reduce symptoms, improving treatment efficacy, and cancer prognosis) [98,101,102,106,107,108,111,115,116,118,119,121,132,138,140,144,147,148]. These perceived and anticipated benefits may be influenced by receiving information on the (health) benefits of favorable lifestyle changes. Receiving *knowledge about lifestyle and the effects on health* was mentioned as a facilitator in qualitative studies [102,108,109,114,124,133,135,136,137,139,140,142].

Some of the included studies examined psychosocial determinants from a theoretical perspective (see Appendix A). For example, some studies studied multiple determinants from Social Cognitive Theory [27,42,62,69,132,150]. This is in line with a previous systematic review showing that lifestyle interventions for cancer survivors have frequently been based on Social Cognitive Theory [151].

### Strengths & Limitations

A strength of this systematic review of the literature is the inclusion of both quantitative and qualitative studies. Including both types of research combines the strengths of both research methods and increases the reliability and credibility of the findings [25]. The results demonstrate the added value of including both types of research, clearly showing the differences and similarities in findings from quantitative vs. qualitative research. For example, numerous additional determinants were retrieved from the qualitative studies in addition to the determinants retrieved from the quantitative studies. These additional determinants obtained from qualitative research reflect the cancer survivors’ perspective (vs. the predominant researcher’s perspective in quantitative studies), which provides additional guidance on how to impact clinical practice and inspires future research.

Another strength is the systematic thorough approach that was applied in this review of the literature. The systematic ordering of the literature per lifestyle behavior provided a detailed overview of the current literature allowing for a specific direction to implications for research and practice. For example, it allows for providing recommendations regarding specific lifestyle behaviors. As each lifestyle behavior is unique, it requires a different health promotion approach. This is illustrated by the observed differences in determinants between lifestyle behaviors.

While interpreting the findings of this review, some limitations should be taken into consideration. Due to the variety in study design of the included studies, we did not conduct a quality assessment. We recommend the reader to incorporate the study characteristics (shown in Appendix A) in interpreting the scientific evidence presented in our systematic review. For example, a large proportion of the included quantitative studies has a cross-sectional study design, whereas either a longitudinal study design or a randomized controlled trial would be preferrable to assess psychosocial determinants of lifestyle changes. In addition, most studies assess lifestyle changes with self-reported data, which could be prone to bias.

This systematic review of the literature provides a wide range of psychosocial determinants of lifestyle change in cancer survivors that can be used to select behavior change techniques and strategies that may be effective in promoting lifestyle change in individual cancer survivors. By matching specific modifiable determinants relevant for this specific patient population to behavior change techniques and strategies, a ‘toolbox’ containing a variety of building blocks (i.e., intervention ingredients) can be created. The Behavior Change Technique Taxonomy [152], the Behavior Change Wheel [153], and Intervention Mapping [154] could be used to translate these psychosocial determinants into personalized interventions. The importance of such personalized interventions (i.e., personalized lifestyle medicine) is widely recognized nowadays [155,156]. Besides psychosocial factors, many other factors (such as environmental factors on the practice and policy level) may influence lifestyle changes after the diagnosis cancer. Although these factors were not within the scope of this systematic review, they do need to be taken into consideration while promoting lifestyle changes after a cancer diagnosis.

While translating these psychosocial determinants into personalized interventions, the definitions of these determinants should be carefully taken into consideration as differences in definitions may lead to different operationalizations in interventions. In the different included studies, as well as in different theories and models of health behavior change, different terminology may be used to describe similar concepts, such as perceived behavioral control (e.g., defined as “the extent to which a person feels able to perform the behavior” in the Theory of Planned Behavior), perceived competence (e.g., defined as “Seek to control the outcome and experience mastery” in Self-Determination Theory), and self-efficacy (e.g., incorporated in Social Cognitive Theory and in the i-change model) [157]. In some cases, similar terminology is used to describe comparable concepts. For example, self-efficacy is defined as “people’s judgements of their ability to cope effectively in different circumstances” according to the Social Cognitive Theory [157], while according to the i-change model, self-efficacy is defined as “a person’s perception of their ability to carry out the behavior” [157].

The (oncology) health care provider could play an important role in identifying the (most important) determinants of lifestyle changes in an individual cancer survivor. But, first and foremost, the qualitative results of this systematic review illustrate the important role that oncology health care providers (e.g., oncologists, surgeons) play in changing lifestyle from the cancer survivors’ perspective. Our qualitative findings showed that *lack of information or advice from health care professionals* and *lack of knowledge on health benefits* were frequently mentioned as barriers to lifestyle changes and that *perceived/anticipated benefits* were frequently mentioned as a facilitator in the qualitative studies. Oncology health care providers can promote lifestyle changes in the areas in which this is advisable for an individual cancer survivor, by providing evidence-based information and advice on the health benefits of lifestyle change. A source health care professionals could use to obtain evidence-based information about the relation between nutrition, physical activity, and body weight and for evidence-based lifestyle and body weight recommendations for cancer survivors is the website of the World Cancer Research Fund (www.wcrf.org, accessed on 31 January 2022). For cancer survivors, it is important that this information and advice is provided by their oncology health care providers, who they trust and perceive as a credible source, which is a behavior change technique in itself [152]. Other behavior change techniques that could be used to influence some of the determinants that were found to be one of the most influential in this review, include promoting *social support* by asking cancer survivors about their opportunities for social support in their direct social environment (e.g., social support they could receive from their partner, family or friends) and by advising on, arranging or providing social support (e.g., advise to find a buddy to exercise with) [152]. In addition, *self-efficacy* could be increased by applying the behavior change techniques goal setting, action planning, graded tasks, (self) monitoring of behavior, and feedback on behavior [152]. Most of these behavior change techniques ((self-)monitoring and feedback on behavior, goal setting, and action planning) were also mentioned as facilitators in the included qualitative studies. These behavior change techniques can be applied by health care professionals during individual counseling sessions which may be supported by digital technology (such as health apps for mobile phones). The use of digital technology may provide a promising means to assist in initiating and maintaining health behavior changes.

While searching for relevant literature for our review, we noticed that we excluded a large amount of quantitative studies (predominantly randomized controlled intervention studies) that did collect the data to be able to study psychosocial determinants of lifestyle changes in cancer survivors, but did not conduct the appropriate analyses to report on determinants of lifestyle changes as this generally was not the primary purpose of these studies. Similarly, we noticed that numerous included quantitative studies typically reported on psychosocial determinants of lifestyle changes using secondary data analyses. In order to further build the evidence base, we recommend to publish such secondary data-analyses in intervention studies that have already collected data on psychosocial determinants. For future intervention studies, it is recommended to, in addition to an effect evaluation, also conduct a process evaluation to gain more insight into (in)effective components of the intervention and mechanisms of behavioral change, and to include psychosocial determinants in data collection and analyses. In addition, large longitudinal observational studies assessing determinants of lifestyle change are valuable means to further build the scientific evidence base. Given the limited amount of included studies on alcohol (n = 4 quantitative; n = 0 qualitative), sun protection (n = 2 quantitative; n = 0 qualitative), and smoking (n = 13 quantitative; n= 4 qualitative) and given the evidence for the positive health effects of making favorable changes in these lifestyle behaviors [11,12,13,14,15,16], future research on psychosocial determinants of these specific lifestyle behaviors is warranted. As almost all of the included studies were either quantitative or qualitative in nature, it would be a valuable addition to conduct more mixed-methods research in this area. Moreover, it would be a valuable addition to conduct studies on psychosocial determinants of lifestyle changes in cancer survivors, using novel techniques, such as Ecological Momentary Assessment, which has the potential of real-life assessment of determinants of lifestyle change.

## 5. Conclusions

This overview of the scientific literature on psychosocial determinants of lifestyle change in cancer survivors showed that a large variety of determinants may influence lifestyle change after cancer diagnosis. For example, at the inter-individual level, a positive association between *social support* and favorable lifestyle changes was found, particularly for changes in physical activity. In addition, *advice or support from health care professionals* and *receiving professional supervision* were mentioned as facilitators of favorable lifestyle changes, whereas *lack of information or advice from health care professionals* was mentioned as a barrier. Psychosocial determinants at the intra-individual level included *self-efficacy*, *psychological complaints (e.g., depression, anxiety, and stress), (lack of) motivation, experienced benefits from healthy lifestyle behaviors, perceived or anticipated benefits of lifestyle change,* and *receiving knowledge about lifestyle and the effects on health*. Findings from this systematic review of the literature demonstrate the important role of oncology healthcare professionals in promoting healthy lifestyle changes in cancer survivors. In addition, findings inform researchers involved in the development of health promotion programs about the methods and strategies they can use to promote healthy lifestyle changes in cancer survivors. Promoting lifestyle change among cancer survivors is expected to have beneficial effects on cancer risk and overall health.

## Figures and Tables

**Figure 1 cancers-14-02026-f001:**
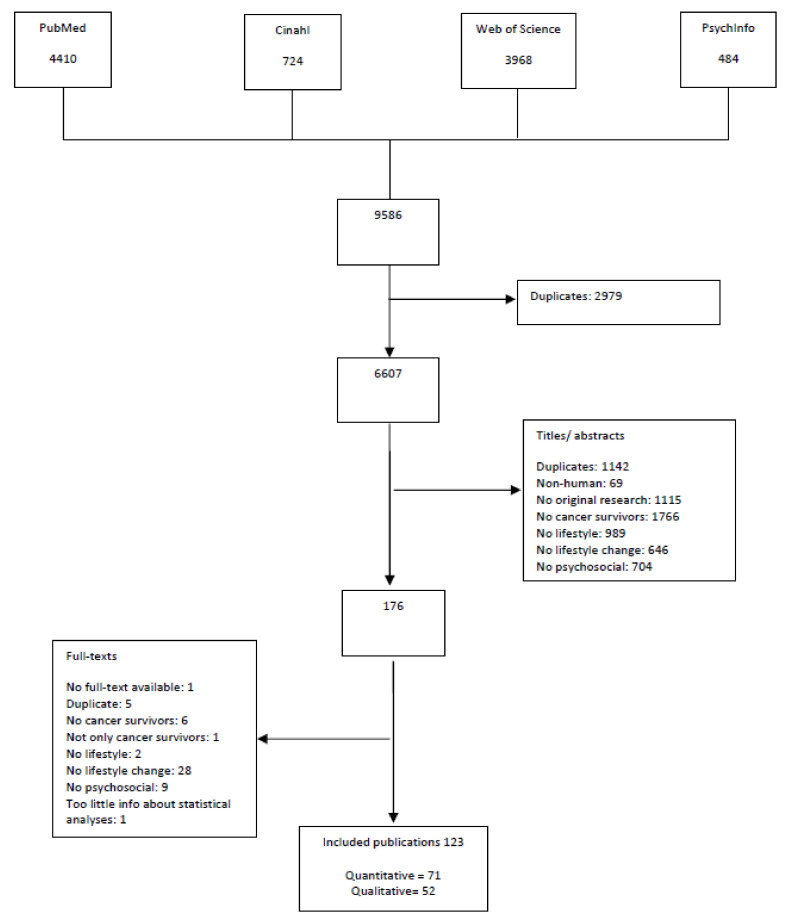
Flow-chart of inclusion and exclusion of publications derived from the database searches.

**Table 1 cancers-14-02026-t001:** Search terms used to select original research on psychosocial determinants of lifestyle changes in cancer survivors.

	Search Terms
Cancer survivors	“Cancer” OR “Cancer patients” OR “Cancer survivors” OR “Neoplasms” OR “Oncology”
Lifestyle	“Lifestyle” OR “Life style”
Physical activity	“Physical activit*” OR “Exercis*” OR “Strength training” OR “Aerobic” OR “Resistance training” OR “Walking” OR “Sitting” OR “Sedentary behaviour” OR “Sedentary behavior”
Diet	“Diet*” OR “Nutrition” OR “Food” OR “Fruit” OR “Vegetable” OR “Meat” OR “Red meat intake” OR “Processed meat” OR “Energy dense food” OR “Fast food” OR “Processed food” OR “Starches” OR “Sugar” OR “Sugary drinks” OR “Sugary drink intake” OR “Fiber intake” OR “Wholegrains”
Smoking	“Smoking” OR “Smoking cessation” OR “Tobacco”
Alcohol consumption	“Alcohol consumption” OR “Alcohol” OR “Alcohol drinking” OR “Ethanol”
Sun protection	“Sunscreen” OR “Sun block” OR “Tanning” OR “Tanning bed”
Change	“Change” OR “Promotion” OR “Behavior change” OR “Modification” OR “Intervention” OR “Program” OR “Trial”
Psychological	“Psycholog*” OR “Psychopathology” OR “Anxiety” OR “Depression” OR “Intrapsychological” OR “Self-efficacy” OR “Selfefficacy” OR “Mastery” OR “Motivation” OR “Coping” OR “Emotion regulation” OR “Personality” OR “Attachment” OR “Trauma” OR “Adverse childhood events” OR “ACE” OR “Resilience” OR “Perceived stress” OR “Worry” OR “Fear” OR “Distress” OR “Mental health” OR “Emotional functioning” OR “Emotional well-being”
Social	“Social” OR “Social support” OR “Social pressure” OR “Socioeconomic status” OR “SES” OR “Educational level” OR “Marital status” OR “Partner” OR “Family” OR “Social environment”

Abbreviations: ACE = Adverse Childhood Events; SES = socio economic status.

**Table 2 cancers-14-02026-t002:** Overview of included quantitative studies on psychosocial determinants of (favorable) lifestyle changes in cancer survivors.

	Physical Activity (n = 45)	Diet (n = 21)	Smoking (n = 12)	Alcohol (n = 4)	Multiple Lifestyle Behaviors (n = 4)	Sun Protection (n = 2)
**Psychosocial Determinant**						
*Socio-demographic*						
Age	Not Significant (NS) [1,2,3,4,5,6,7] **Older age* & physically inactivity [8] **Younger age* & increased exercise frequency [9] **Age* differed significantly between trajectory groups of the waitlist group [10]	NS [5,8,11] **Younger age* & (favorable) dietary changes [12,13,14] **Older age* & favorable dietary changes [6]	NS [15,16,17,18]* *Older age* & lower likelihood of continued smoking [19] **Older age* & smoking cessation [20] **Younger age* & more likely to continue smoking [21]	NS [3,11]	NS [22,23,24]	**Age > 55* & increased sun-safe behavior [6]
Sex/gender	NS [10] **Gender* differed across classes: males more likely to be high and sustained sedentary over time; women more likely to be increasing sedentary [7]		NS [15,16,17,18,21] **Females* less likely to quit smoking [20]		NS [22] **Females* & less positive changes in substance use (alcohol and smoking) [22]	
Race/ethnicity	NS [1,4]		NS [15,16,17,18]		NS [23,24]	
Educational level	NS [1,5,6,7,9,25] NS for Moderate to Vigorous Physical Activity (MVPA) [4] **Higher educational level* & more likely to change towards being physically inactive post-diagnosis [8] **Higher educational level* & increase in physical activity vs. no increase among lower educational level [26] **Higher educational level* & more likely to be high maintainers or high decreasers of sedentary behavior, vs. low maintainers [4]	NS [5,6,8,13,14,25] **Higher education level* & (favorable) dietary changes [11,27]	NS [16,17,19,21,28] **Lower educational level* & lower long-term cessation rates [15]	NS [28] *Temporary decreasers were more likely to have a *higher education level* vs. medium temporary decreasers vs. low maintainers [11]	NS [22,23] *Higher *educational level* & positive changes in physical activity or diet [24]	NS [6]
Employment status	NS [1,5,6,8] **Employment* differed across classes (those increasing sedentary behavior over time were most often employed) [7]	NS [5] **Being employed* & increase in fiber intake [6]			NS [22]	NS [6]
Job position	**Higher occupational positions* & less improvement in moderate physical activity [5]	NS [5]				
Marital status	NS [1,3,5,6,8]	NS [5,6,8,13]	NS [15,19,21,28] #*Married/partnered* more likely to be abstinent [16]	NS [3,28]	NS [22,23]	NS [6]
Social class	*Working *occupational class* more likely to increase physical activity compared to managerial and professional class [3]	**Higher social class* & favorable dietary changes [12]		NS [3]		
Cohabitation/living alone		**Living alone* & favorable dietary changes [12]				
Income	*Higher income & more likely to be high decreaser or medium decreaser of physical activity vs. low maintainer [4]	NS [14] **Higher household income* & favorable dietary changes [4]	NS [15,21]		NS [22]	
Smokers in household			**Smoking household member* & higher likelihood continued smoking [19] **Second-hand smoking at home* & being indecisive for abstinence [21]			
*Inter-individual*						
Social support	NS [4,29], for baseline to 6 months [32] #Social support & increasing physical activity [30] *More *social support* [1,31] from family [32], friends [34], or family [40] and friends [32] & (greater) increase in physical activity**Social support* & increased exercise from baseline to 6 months follow-up, but not at 3-month follow-up [9]	NS [30] *Lower friend support for eating habits-discouragement & improvements in diet [33] *Greater social support & increase in fruit and vegetable intake [4]	NS [18,34]	NS [11]	**Social support* & positive lifestyle change [23]	
Social modeling	*Increases in *social modeling* & increases in physical activity and decreases in sedentary time [31]					
Exercise role models	NS [29]					
Role model	*Contact (vs. no contact) with a *role model &* increase in exercise levels [35]					
Social constraints					NS [23]	
Social smoking environment			*Having a spouse who did not smoke, and having fewer peers who smoked & higher likelihood to quit [34]			
Second-hand smoke exposure at home			*Exposed to send-hand smoke at home & smoking over time [21]			
*Intra-individual*						
Depressive symptoms	NS [4,8,30,32,36] **Higher levels of depressive symptoms* & less likely to remain consistently sufficiently active [2] *Depression perceived barrier [30]	NS [4,8,30,37]; NS for those with better fruit and vegetable and fiber intake at baseline [38] *For those with less fat consumption at baseline, increase in fat intake, depressive symptoms were not associated with decreases in the first year, whereas it was associated with the increase between year 1 and 4 [38]	NS [16,18,21] *Depression & higher likelihood of continued smoking [19] * Patients with *depression symptoms* reported significantly lower abstinence rates [39] **Depression* & relapse after quitting [15]	NS [11]	NS [40]	
Anxiety symptoms	NS [8] **Higher anxiety* & less likely to increase physical activity [36]	NS [8,37]	NS [18,21] **Lower anxiety* & abstinence [17]		* *Higher anxiety* & unhealthy lifestyle [40]	
Psychological distress		NS [18] **Higher psychological distress* & initiating dietary changes [14] **Decrease in psychological distress* & dietary changes [14]	**Lower psychological distress* & abstinence rates [17]			
Emotional distress	**Higher emotional distress* & decrease in physical activity [41]		NS [42]	**Higher emotional distress* & increased alcohol consumption [41]		NS [41]
Stressful life events	NS [30]	NS [30] **Greater number of stressful events* & initiating dietary changes [14]				
Life stress	NS [36]					
Perceived stress	**Perceived stress* as barrier [30]	**Lower perceived stress* & smoking abstinence [17]	NS [43]			
Cancer-related stress		**Higher stress* & greater decrease in fruit and vegetable intake in first 6 months after diagnosis [44] **Higher stress* & greater increase in fruit and vegetable intake 12 months post-diagnosis [44]			NS [22]	
Traumatic stressor response					NS [23]	
Cancer-related intrusions					**Cancer-related intrusions* & positive lifestyle change [23]	
Cancer-related avoidance					NS [23]	
Fear of exercise	NS [29]					
Fear of recurrence	NS [30] **Higher fear of cancer recurrence* & reduced physical activity [41]	NS [30,41]	NS [16]	**Higher fear or recurrence* & increased alcohol consumption [41]		NS [41]
Fatigue (vitality)	NS [28] **Less fatigue* & increase in exercise [30] **Higher levels of fatigue* at baseline & less likely to remain consistently sufficiently active [2]**Baseline fatigue* & physical activity maintenance [45]	# *Less fatigue (greater vitality)* & dietary changes [30]	NS [16,18]			
Mood	**Lower mood disturbance* at baseline & low and sustained sedentary behavior over time [7]		NS [18]			
Anger			NS [18]			
Confusion			NS [18]			
Vigor			*Higher *vigor* for continuous abstainers than relapsers [18]			
Dispositional optimism	NS [4]	**Higher dispositional optimism* & higher fruit and vegetable intake [4]		NS [11]	**Dispositional optimism* & positive lifestyle change [23]	
Contemporary life stress		NS [46]				
Sexual activity, sexual functioning		NS [27]				
Satisfaction with sexual functioning	NS [30]	NS [30]				
Body satisfaction	NS [30]	NS [30]				
Health related quality of life	**Poor health related quality of life* on two or more domains & exercising less [47] **Higher mental and physical component scores* & increase in physical activity [32]	NS [47] **Lower general quality of life, lower cognitive functioning, lower levels of emotional functioning*, & dietary changes [27]				
Perceived mental health status	**Better mental health status* & increased exercise frequency from baseline to 3 months follow-up, but not at 6 months follow-up [9] *Perceived *reduced mental function* as barrier [30]					
Health awareness	**Higher health awareness* & less physical activity [48]	NS [48]				
Meaning of cancer	NS [48]	NS [48]				
Survivor concerns		NS [37]				
Cancer-specific concerns	NS for breast cancer survivors [49] *For prostate cancer survivors, *cancer-specific concern* of ‘activities limited by urination’ & lesser increases in physical activity [49]					
Appearance concerns	NS [48]	NS [48]				
Body change concerns	* *Higher body change concerns* & less physical activity [48]	NS [48]				
Life interferences	* *Higher life interferences* & less physical activity [48]	NS [48]				
Worry	**Worry* & less physical activity [48]	NS [48]				
Cancer worry	**Higher levels of cancer worry* & more likely to remain consistently sufficiently active [2]					
Illness representations (timeline acute/chronic, timeline cyclical, consequences, personal control, treatment control, illness coherence and emotional representations)	NS for *illness coherence* or *consequences* [50] NS for *timeline acute/chronic, timeline cyclical, consequences, illness coherence and emotional representations* [25] **Lower personal control* & decrease in exercise [50] **Lower emotional representations &* decrease in exercise [50] **Lower illness identity, higher personal control, higher treatment control* & increase in physical activity [25]	**Higher personal control* & healthier changes [25] **Higher negative emotional representations* & healthier changes [25]				
Self-efficacy	NS [7,9,25,45,51,52,53,54,55] NS in the control group [56] **Higher self-efficacy* & higher (increase in) physical activity [31,33,57,58,59] (in the intervention group [56]), being sufficiently active [60] **Lower self-efficacy* & decreasers [50], lower physical activity [10]	NS [25] **Higher (changes in) self-efficacy* & (favorable) dietary changes [15,33,61,62] #*Higher self-efficacy* & target fruit and vegetable intake [37]	NS [16,42] **Higher self-efficacy* & quit attempts [15] **Higher self-efficacy* & continuous abstainers [18] **Lower self-efficacy* for not smoking & still smoking over time [21]			
Task self-efficacy	NS [29,53] *Increase in *task self-efficacy* & improved physical activity [63]					
Barriers self-efficacy	NS [59] *Changes in *barrier self-efficacy* & changes in steps per day in the intervention group [64]. **Increased barrier self-efficacy* & improved vigorous physical activity [63] #Increase in *barrier self-efficacy* & increase in walking and decrease in sitting time [31]					
	*Improvements in *barriers self-efficacy* mediated intervention effect on physical activity maintenance [29]					
Relapse self-efficacy	*Changes in *relapse self-efficacy* & changes in steps per day, in the intervention group [64]					
Maintenance self-efficacy	NS [51] *Higher (change in) self-efficacy & increase in physical activity during intervention, but not at 10 week follow-up [65]					
Perceived behavioural control		*Lower external locus of control & dietary changes [27]				
Positive outcome expectations	NS [59]					
Negative outcome expectations	NS [59]					
Outcome expectations	NS [29,53,57] **Exercise outcome expectancy* (beliefs that exercise has beneficial consequences) & increased exercise from baseline to 6 months follow-up, but not at 3 month-follow-up [9]					
Sociostructural factors	**Reductions in motivation* & improved follow-up vigorous physical activity [63]					
Locus of control		NS [37]				
Stage of change	**Higher stage of change* & increased exercise since diagnosis [25]	NS [37] **Higher stage of change* & healthier eating since diagnosis [25]	**Lower readiness to change* & continuous smoker vs. quitter [15] **Higher readiness to change* & quit attempts [15], less likely to relapse [18] #*Higher stage of change* & smoking cessation [66] **Quit motivation* & smoking cessation [42]			
(Exercise) Processes of change (behavioral and cognitive)	NS for *behavioral processes* [31] NS for *cognitive processes* [32] **Cognitive processes* & increasing sedentary time [31] **Behavioral processes* & greater change in physical activity at 6 months and 12 months [32] **Behavioral processes* & greater odds of being sufficiently active at follow-up [60]					
Change processes	*Behavioral processes of change* & increase in pounds lifted for leg exercises, but not for arm exercises [52]					
Perceived access (to healthy eating; to exercise)	**Higher perceived access to exercise* & increased physical activity [34]	NS [33]				
Perceived neighborhood safety	NS [34]	NS [33]				
Change in barriers		*Perceiving less barriers & diet quality [62]				
Healthy food beliefs		NS [37]				
Behavioral capabilities		NS [37]				
Difficulty finding fruit and vegetables in the neighborhood		NS [37]				
Difficulty eating fruit and vegetables as snack		NS [37]				
Taste and snack preferences for fruit and vegetables		**Improved taste/snack preferences for fruit and vegetables* & increase in fruit and vegetable intake [37]				
Family opinions on fruit and vegetables		NS [37]				
Cancer coping style	NS [67]	*Fatalists (vs. fighting spirits) & increase in fruit and vegetable intake [67]				
Fatalism			NS [42]			
Coping behaviors to resist smoking			NS [18]			
Stress coping			NS [17]			
Risk perception			NS [16,42]			
Cancer threat appraisal	NS [36]					
Decisional balance: Pros and cons	NS [52,60] **Higher decisional balance pros* and *lower decisional balance cons* & greater physical activity at 6 months, but not at 12 months [32]		**Cons* & smoking cessation at 3 months [42]			
Pain			NS [16]			
Benefit finding					*Benefit finding & increase in lifestyle behavior [22]	
Motivational regulation (self-determined motivation, amotivation, external regulation and introjected regulation)	*Increase in self-determined *motivation* & increase in moderate to vigorous physical activity [68] NS: other subscales [68]					
Motivation	NS [64]					
Motivational processes (instrumental attitudes, affective attitudes, perceived capability and perceived opportunity)	*Higher perceived opportunity* & greater changes in physical activity [69] Other subscales NS [69]					
Behavioral regulations (exercise action and coping plans, and social support)	NS [69]					
Reflexive processes (anticipated regret, habit, exercise identity, exercise obligation, and regulation of alternatives)	NS [69]					
Somatization	*Increased *somatization* increased & less likely to increase physical activity [36]					
Belief that exercise has a negative impact on cancer	*Main effect NS, but decreasers were more concerned about the negative impact of exercise on cancer than increasers [50]					
Perceived benefits of exercise	NS [50]					
Perceived barriers (of exercise)	NS [50,63] **Perceived barriers* & increased exercise frequency from baseline to 3 months, but not at 6 months follow-up [9] **Reductions in barriers* & greater physical activity [53]					
Barrier interference	**Barrier inference* mediator of intervention effect on physical activity [29]					
Perceptions of physical activity	NS for maintenance of physical activity after diagnosis [70] For patients not meeting guidelines before diagnosis, *perceptions of physical activity improving quality of life and overall survival* & increased physical activity after diagnosis [70]					
Physical activity enjoyment	NS [29] *Increase in *physical activity enjoyment* significantly predicted physical activity at post-intervention [71]					
Coping planning	NS [72]					
Action planning	**Action planning* & MVPA [51] *Greater *action planning* & maintenance of exercise for more than 6 months [72]					
Intention	**Intention* & MVPA [51]					
Self-leadership (behavior awareness and volition, task motivation, and constructive cognition)	**Higher self-leadership in the subscales: behavior awareness and volition, task motivation, and constructive cognition* & maintenance of moderate exercise during 6 months [38]					

* = *p* < 0.05; # = Trend; *p*-value between 0.05 and 0.10; NS = Not (statistically) Significant; MVPA = Moderate to Vigorous Physical Activity.

**Table 3 cancers-14-02026-t003:** Summary table of included qualitative studies on psychosocial determinants of lifestyle change in cancer survivors (n = 52).

	Physical Activity (n = 26)	Diet (n = 9)	Smoking (n = 4)	Multiple Lifestyle Behaviors (n = 13)
**Barriers**				
* **Sociodemographic** *				
Work-related factors	[1,2,3]	[4]	[5]	
Financial constraints	[6,7,8,9,10]	[4,11]		[12,13,14]
Ageing	[1,10,15,16,17]			[12]
Poor weather conditions	[1,3,6,7,8,15,17,18,19,20,21,22]			[13,23,24]
Environmental factors (e.g., poor infrastructure)	[10,19,22]			[13]
* **Inter-individual** *				
Lack of information/advice from health care professionals	[6,8,9,10,17]	[11,25,26]		[12,13,14,27,28,29]
Lack of trustworthy lifestyle information				[13,27,30,31]
Lack of knowledge	[7,8,9,10,16,32]			[30,33]
Lack of discussion about lifestyle with health care professionals		[34]	[35,36]	
Health care providers authoritarian approach				[14,27,37]
Resistance from family members to dietary changes		[4,38]		
Poor support and understanding from family members				[29]
Living alone/not having a partner		[4]		[23,24,31,39]
Practicing alone	[2]			
Difficulties with breaking (cultural) dietary patterns		[4,40]		[30]
Difficulties breaking old and forming new habits				[12,27,29,37]
Social isolation/feeling isolated	[19,20]	[40]	[5]	[29]
Not wanting to bother the host with dietary restrictions		[11,40]		
Perceiving smoking as a social norm and as a tool for communication and connecting with friends			[41]	
Feeling impolite or embarrassed to reject food prepared by others/a cigarette from a friend		[40]	[41]	
Dilemma between staying on a healthy diet and maintaining harmony with others		[40]		
Residing with other smokers			[5]	
Social pressure (e.g., pressure to stop smoking from relatives)			[36]	[27]
Timing of the intervention (during radiotherapy)		[11]		
Unfavorable lifestyle and lack of lifestyle change in social environment				[14,29]
Difficulties in shopping for food		[11]		
Specific social events		[11,40]		[14]
Unexpected (major) life events (e.g., serious illness, death)	[22]			[29]
Belief that weight loss is a positive health outcome of cancer		[34]		
Not being able to consume foods that one typically consumed interferes with normative expectations		[34]		
Shift in domestic food dynamics: disruption of traditional gender roles		[42]		
Difficulties resuming life roles				[27]
Passive role in food decisions/preparation		[42]		
Negotiating (with partner) to find a balance between dietary regimens and living an enjoyable life		[42]		
Issues with facilities or resources (e.g., proximity/access to facilities)	[1,3,6,8,10,16,43]			
Lack of program flexibility (e.g., unchallenging exercise regimes)	[6]			[23]
Competing time demands (e.g., balancing motherhood with healthy lifestyle; attending smoking cessation services)	[1,2,6,7,8,10,15,17,19,20,21,22,32,43,44]		[36]	[13,23,24,37]
Safety issues	[6,8,19]			
Grief about inability to engage in normal group sport activities	[20]			
Difficulties maintaining change after end of intervention/post-program lack of external encouragement	[19]			[29]
Feeling no need to exercise because of regular medical checkups	[17]			[33]
Current practice in smoking cessation services			[36]	
Obesity-related social stigma				[13]
* **Intra-individual** *				
Physical complaints/treatment side effects	[1,2,6,7,8,9,10,15,16,17,19,20,21,22,32,43,44,45,46,47,48,49,50,51]			[12,13,14,23,24,27,29,30,39]
Lack of information about diet and cancer		[25,42,52]		
Perceiving no need for lifestyle change		[4,25]		[12,13,30]
Misperceptions about recommendations/guidelines not applicable	[17]			[29]
Overestimation of own levels of physical activity	[17]			
Not being too concerned about effects of smoking			[5]	
Beliefs about (the cause of) cancer being unrelated to lifestyle		[4]		[12,27,30]
Concurrent health concerns (e.g., Crohn’s disease)		[4]		
Feeling restricted/limited to eat specific foods		[11]		[12]
Need for control/autonomy over lifestyle choices		[11,26]		[27]
Frustration and embarrassment to eat with others because of bodily changes caused by cancer and cancer treatment		[34]		
Lack of interest in food		[42]		
Lack of skills		[42]		
Changed body image & inconvenience and worries due to using a prosthesis	[32,43]			
Concerns/anxiety about exercising	[1,2,6,9,20,49]			
Lack of knowledge and limited perceptions (e.g., on smoking cessation and health consequences)	[6,16,17,18]		[36,41]	
Lack of motivation	[1,2,6,7,8,10,15,16,17,19,20,21,22,51]	[11]		[29,33]
Not being the sporty type	[1,2,7,17,22,51]			
Low self-efficacy	[8,9,46]			[30,33,37]
Not enjoying healthy behaviors	[1,10,15,48,49]			[13,23,24]
Enjoyment of unhealthy behaviors				[27]
Being unfamiliar with healthy products and digital technology (e.g, m-health)	[2]			[12,24]
Unclear about feasible activities	[20]			
Lack of sport equipment	[20]			
Concerns/fears related to symptoms (body esteem, colostomy bag leakage, and accidents)	[15,18]			[24,33]
Not prioritizing physical activity	[16]			
Counterintuitive approach	[2]			
Inconvenience/Eating unhealthy foods for convenience	[6]			[29]
Eating unhealthy foods for palatability				[29]
Preoccupied with dealing with cancer	[51]			
Uncertainty about benefits of lifestyle in relation to cancer and health/Not perceiving any benefits of lifestyle change (e.g., smoking cessation)			[41]	[12,27,30,33]
Physical dependence/Nicotine dependency			[41]	[27]
The stress of being away from home (in hospital)			[5]	
Experiencing a strong desire to smoke			[41]	
Difficulties to quit			[35,36,41]	
Lack of willpower			[5]	
Marijuana use			[5]	
Uncertainty on how to approach quitting			[35]	
Poor/uncertain disease prognosis			[35]	[27]
Negative views about current smoking cessation services			[36]	
Coping with (emotional di)stress trough unhealthy behaviors			[36]	[27,29,39]
Desire for personal choice over smoking behavior			[36]	
Desire to move on from cancer diagnosis and treatment				[12]
Self-monitoring perceived as discouraging when not meeting goal				[31]
Inner conflicts				[37]
Passive surrender to avoid disappointment from unsuccessful attempt to change lifestyle				[37]
Psychological complaints (e.g., low mood, depression, stress, anxiety)	[6,9,15,19,20,21,22,44,45,51]		[5]	[13,14,27,39]
Feeling hungry				[29]
Desire to enjoy life and not having to constantly monitor lifestyle				[14,33]
**Facilitators**				
* **Sociodemographic** *				
Being retired		[4]		
Ageing				[12]
Affordability/smoking cessation saves money	[6,9]		[36,41]	
Environmental factors (e.g., proper infrastructure)	[10]			[13]
Good weather				[13]
* **Inter-individual** *				
Social support (e.g., from partners and family members)	[2,3,7,8,9,10,15,16,17,18,19,22,32,43,44,46,47,49]	[4,11,25,26,38]	[5,35,41]	[12,13,14,23,24,27,29,31]
Advice/support from health care professionals	[7,16,17,32,47,51]	[11,25,40,42]	[5,35]	[12,23,27,29,31,37]
Credible source				[24,31]
Receiving professional supervision/Prior education on addictions and withdrawal through occupational interventions	[2,7,9,15,18,32,45,47,48,49,50]		[5]	
Patient engagement				[27]
Greater priority for healthy eating due to diagnosis		[38]		
Sharing cooking responsibilities		[4]		
Being responsible for cooking for family members				[14,29]
Living alone		[4]	[5]	
Familiarity with healthy eating tradition		[4]		
Prior knowledge and experience with healthy products		[11]		
Believing that weight loss is desirable		[34]		
Partner adjustment in role functioning regarding food provision		[42]		
Medical justification of dietary changes (to others)		[42]		
Using adaptive strategies in interpersonal contexts		[40]		
Accessibility of facilities/resources	[6,7,20,44]			[13,14]
External accountability (Feeling personally accountable to the coach)	[10,16]			[23,29,31]
Avoiding/reducing isolation	[15,48]			
Benefits of being/exercising with fellow sufferers	[3,18,43,44,47,49,50]			
Enjoyment of group exercises				[13]
Routine & structure	[2,7,32,44,45,48,49]			[23,29,31]
Commitment	[8,19,32,46,49]			[23]
Printed intervention components	[51]			
Being physically active together helps coping with cancer	[7]			
Having a pet (e.g., owning a dog)	[7]			[13]
Social norms	[8]			
Tailored step goals (set by researchers)—Tailored, individualized exercises	[19]			[23]
Monitoring/visualization of progress/Intervention raises awareness of health behaviors and outcomes	[19]			[31]
Exercising in public gym provides a sense of normalcy and health	[49]			
Getting asked to exercise	[20]			
Being away from home			[5]	
Social unacceptability of smoking			[35,36]	
Caring responsibilities			[36]	
Use of cessation services			[36]	
Feelings of responsibility and gratitude toward family members				[28]
Meal provisioning				[23]
* **Intra-individual** *				
Cancer diagnosis as wake up call—as initial motivating factor		[25,38]	[5]	[12,13,24,27,31]
Knowledge (about lifestyle and effects on health)	[6,22,44,49,50]	[25,40]	[5,35,41]	[13,29]
Fear of recurrence & perceiving that lifestyle change may prevent recurrence	[3]	[4,25,26,40,52]		[14,27,28,29,37]
Perceived/anticipated benefits of lifestyle change: to improve health, wellbeing, reduce symptoms, improving treatment efficacy & cancer prognosis	[8,10,15,20,32,46,51]	[11,25,34,52]	[41]	[12,13,14,24,28,30]
Lifestyle change as active coping strategy: doing something to gain a sense of control		[11,25,26,34,52]		[31]
Experienced benefits from healthy behaviors (e.g., improved mental wellbeing; help process negative thoughts and feelings)	[2,7,8,9,10,15,18,19,20,22,32,43,44,45,46,47,48,49,50,51]	[26,40]		[12,13,28,29,30,37]
Personal/internal motivation and commitment	[3,47,48,49,51]	[38,40]		[24,29,31,33]
Food as a source of comfort		[52]		
Concurrent health concerns already requiring dietary changes (e.g., diabetes)		[4]		
Interest and knowledge in food and cooking		[11,42]		
Positive experience of novel dietary knowledge and habits		[11]		
Recipes and meal suggestions		[11]		
Small dietary adjustments perceived as easy		[11]		
Shift in meaning of healthy lifestyle behaviors after diagnosis (focus on health)		[26]		[37]
Wanting to return to pre-diagnosis normality		[26]		
Relaxing diet rules (having occasional treats)		[42]		
Having multiple exercise options to choose/Benefit of trying different types of activities to maintain motivation	[6]			[14]
Enjoyment of healthy lifestyle behaviors	[3,6,7,8,15,43,47]			[12,24,31]
Self-efficacy	[10,19,32,46,47,51]			[29,30]
Goal setting/action planning	[7,10,22,51]			[13,14,23,31,37]
Pride	[8,44,47]			
Improved wellbeing leading to prioritizing physical activity	[47]			
Physical activity provides a purpose	[44]			
No self-pity, looking forward	[44]			
Focus on health/living, distraction from illness	[15,44]			[27]
Regaining trust in own body	[44]			
Re-gaining control/being able to do something	[3,7,9,15,18,32,43,44]			
Previous exercise experience	[2,18]			
Objective indicators of improvement	[18]			
(Self-)Monitoring and feedback on behavior	[2,10,15,18,32,48,49,50,51]			[13,29,31]
Habit formation	[2,8,22]			[29]
Openness to reframing attitudes about lifestyle modification	[8]			[37]
Restoring normalcy/Returning to normal life	[20,32,46,51]			[27]
Learning new skills	[49]			
Music	[50]			
Self-challenge	[50]			
Negative reinforcers (e.g, feeling guilty for not exercising)	[10]			
Intrinsic rewards (e.g, feeling good after meeting challenges)	[10]			
Fitness being part of self-identity	[51]			
Positive coping strategies	[22]			
Feelings of empowerment and independence	[44]			
Not wanting to compromise their treatment			[5]	
Being too unwell to smoke because of the side effects of radiotherapy			[5]	
Treatment and its associated side effects			[5]	
Fear of being discovered by the exhaled carbon monoxide readings			[5]	
Willpower			[5,36]	
Cessation aids			[5]	
Removing the association between alcohol and smoking			[5]	
Individual decision to quit			[35]	
Harm recognition			[35]	
Accomplishment in quitting			[35]	
Positive self-talk				[27]
Lifestyle changes complementing existing diet				[12]
Autonomy				[12,31]
Acceptance				[27]
Increased self-awareness/mindfulness				[23]
Experienced discomforts from unhealthy behaviors				[28]
Strength and resilience				[27]
Religion/spirituality				[30]
Intention				[30]
Pro-actively searching for information about lifestyle and health				[28,30]
Rewards				[13]
Portion control				[29]
Skill-building, e.g., in food preparation and meal planning				[14]
Body image				[14]
Engaging children in healthy lifestyle behaviors				[37]
Having a more self-compassionate perspective				[37]

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
