# Peer review of "Psychosocial Determinants of Lifestyle Change after a Cancer Diagnosis: A Systematic Review of the Literature"

_cancers, 2022, doi:10.3390/cancers14082026_

Round 1

Reviewer 1 Report

Abstract:

  • Missing time frame for search and information on inclusion criteria

Introduction:

  • Title is referring to lifestyle changes after a cancer diagnosis. The introduction (l. 41-43) seems to indicate a focus on survivorship. Perhaps this should be reflected in the title?
  • Lines 66-71: The rationale is well-articulated. However, the authors already locate the barriers to lifestyle changes on the individual level, assuming that specific behaviours and cancer survivors need to be targeted. I wonder if a psychosocial approach also needs to account for the constraints in which behaviours are situated. In other words, I encourage the authors to think beyond the individual and offer insights on how policy and practice can support lifestyle changes by addressing social inequality and structural constraints on lifestyle choices.

Methods:

  • Line 91: If the full texts were divided among 3 researchers for screening, where did the inconsistencies come from?
  • Line 100: Do you mean ‘consist’ rather than ‘exist’?
  • Line 101: ‘did not’ rather than ‘didn’t’. Do not use contractions. Also later line 764.
  • Was there a risk of bias assessment? I miss here some commentary on the methodological strengths and weaknesses of the reviewed studies. I see that this comes a bit later in the discussion (l. 715 -723), however, this is somewhat unsatisfactory. I agree, that a risk of bias assessment should not lead to the exclusion of studies from review, but, if done well, it can highlight challenges common for research with this population, and possible examples of good practice, or suggestions for improvements to data infrastructure.

Results:

  • Section 3.1 is really hard to read. Consider presenting this information visually, e.g., in figures or in a summary table.
  • I am not sure, why studies are presented along a distinction between quantitative and qualitative work. I suggest integrating those wherever possible.
  • I would also think it important to differentiate populations further, e.g., along age (i.e., AYAs and childhood cancer survivors and other populations) and survivorship state (e.g., in remission, treatment ongoing, etc.). Also, in the presentation of results that comes next, findings from studies are too detached from the local (i.e., region) and clinical (i.e., diagnosis) context which is unfortunate.
  • I understand the need to report on sample sizes, but the way this is done in lines 125 and 143 does not add much value.
  • Define key concepts (i) socio-demographic; (ii) inter-individual; (iii) intra-individual determinants. How are the boundaries drawn around these categorisations? I am not sure that ‘determinant’ is a good term to use either, given that these are often not really ‘determining’ anything conclusively. Perhaps, a less deterministic terminology, such ‘as factors’ is more appropriate?
  • For ease of reading, consider using percentages, e.g. '8 out of 10 studies (i.e., 80%)  reported…'
  • What does ‘higher stage of change’ (l. 273) mean?
  • I understand that this might be a matter of preference: to me the structure along lifestyle behaviours feels unfortunate and repetitive. There seems to be overlap between the ‘determinants’ that cuts across different behaviours. To me it would make sense to synthesize the results along the determinants as the higher order heading, and within those tease out differences across behaviours where applicable. Given that ‘determinants’ are the key interest of the manuscript, this would provide the reader a better opportunity to see at a glance, what the determinants are and how they play out across different lifestyle components. This would also remove the need for a separate subheading on ‘multiple lifestyle behaviours’ (l. 485) which sits a bit awkwardly relative to the previous subheadings.

Discussion:

  • Perhaps this could even be addressed earlier in the manuscript, but I am also missing some comments on the conceptual framework on which the focus on lifestyle behaviours rests. Did the reviewed studies and interventions employ theory to guide their studies? What are the assumptions on which the present systematic review is based?
  • Line 732: I don’t think this is what ‘personalised medicine’ means. At least, I am only familiar with this term in the context of targeting treatments based on genetic information on tumours.
  • The emphasis on the role of healthcare professional comes a little bit unexpected given that nothing in the presentation of results touched on the specific stakeholders in the lifestyle change process of cancer survivors.

Written expression:

  • Signposting, transitions, and presentation of an overall narrative would go a long way in making the results section more compelling. How can these granular findings be synthesized into mid-level insights that are actionable? The results stand out, particularly, in contrast to the well-written and engaging introduction. 

General comments:

  • Unfortunately, I was unable to access the supplementary material, tables and figures that are part of the submission. Hence, my comments are restricted to the main manuscript.
  • The process for the review outlined by the authors is transparent and appears to be thorough.

Reviewer 2 Report

  1. In the first paragraph of ‘introduction’ section in the line-“Favorable lifestyle changes, such as increasing physical activity or smoking cessation, may optimize these health outcomes and increase health-related quality of life among cancer survivors.” In addition, it would benefit from a more in depth presentation regarding life style with NK cell function-e.g. healthy life style increase NK cell activity against cancer. (Deng X, et al. biomedicine 2021. doi: 10.3390/biomedicines9050557.).
  2. In the figure, the articles numbers from “Web of Science” should be indicated.
  3. Please correct the style of references, particularly in page 1 to 2 of the manuscript.

Reviewer 3 Report

This is a systematic review of a very complex topic (the interaction between various patient factors and lifestyle modification after cancer diagnosis).  The methods for the systematic review are well explained and consistent with the accepted norms for such reviews.  The authors have done an exhaustive review of both qualitative and quantitative studies.  They have identified useful themes in terms of ways to improve adherence to lifestyle changes after cancer diagnosis.  

Some of the limitations of the paper are dealt with in the relevant section of the paper.  The use of a variety of methods in the various papers and the inability to perform a quality assessment on the papers limits the ability of the authors to add the next step of meta-analysis of the quantitative papers' data.  I think this is well explained, and considering the nature of the paper and the inclusion of qualitative papers as well as quantitative, the conclusions are correctly presented as thematic rather than strictly data driven.  

I wondered about the ways of enhancing self-efficacy mentioned in the paper, and the potential role of personal phone apps in this regard.  The themes identified in the paper could certainly support the use of these apps which are designed to encourage lifestyle change and hold patients accountable.  Combining that with a recommendation from an oncologist or other health professional could be powerful.
